# Production of Recombinant Alpha-Synuclein: Still No Standardized Protocol in Sight

**DOI:** 10.3390/biom12020324

**Published:** 2022-02-18

**Authors:** Mohammed Al-Azzani, Annekatrin König, Tiago Fleming Outeiro

**Affiliations:** 1Department of Experimental Neurodegeneration, Center for Biostructural Imaging of Neurodegeneration, University Medical Center Göttingen, Waldweg 33, 37073 Göttingen, Germany; mohammed.alazzani@med.uni-goettingen.de (M.A.-A.); annekatrin.koenig@med.uni-goettingen.de (A.K.); 2Max Planck Institute for Experimental Medicine, 37075 Göttingen, Germany; 3Translational and Clinical Research Institute, Faculty of Medical Sciences, Newcastle University, Newcastle upon Tyne NE1 7RU, UK; 4Scientific Employee with an Honorary Contract at German Center for Neurodegenerative Diseases (DZNE), 37075 Göttingen, Germany

**Keywords:** alpha-synuclein, Parkinson’s disease, aggregation, synucleinopathies, amyloid

## Abstract

Synucleinopathies are a group of neurodegenerative diseases, characterized by the abnormal accumulation of the protein alpha-synuclein (aSyn). aSyn is an intrinsically disordered protein that can adopt different aggregation states, some of which may be associated with disease. Therefore, understanding the transitions between such aggregation states may be essential for deciphering the molecular underpinnings underlying synucleinopathies. Recombinant aSyn is routinely produced and purified from *E. coli* in many laboratories, and in vitro preparations of aSyn aggregated species became central for modeling neurodegeneration in cell and animal models. Thus, reproducibility and reliability of such studies largely depends on the purity and homogeneity of aSyn preparations across batches and between laboratories. A variety of different methods are in use to produce and purify aSyn, which we review in this commentary. We also show how extraction buffer composition can affect aSyn aggregation, emphasizing the importance of standardizing protocols to ensure reproducibility between different laboratories and studies, which are essential for advancing the field.

## 1. aSyn Species Are Widely Used in PD Research

Alpha-Synuclein (aSyn) is a protein found in nerve terminals and in the nuclei in brain cells [1,2,3], although it is also present in red blood cells and other peripheral tissues including gastrointestinal tract and skin [4,5,6,7,8,9]. It is the main component of Lewy bodies and Lewy neurites, pathological hallmarks of Parkinson´s disease (PD) and related synucleinopathies. aSyn is considered to be intrinsically disordered in its native state [4,10,11,12]. Soluble monomeric aSyn assembles into oligomeric species of different sizes and shapes. Oligomers transform into stable higher order assemblies rich in beta-sheet structures that bind certain fluorescent dyes such as Thioflavin T. The end-products of the assembly process are highly heterogenous with regards to their size, shape and availability to bind certain dyes [13,14] and, despite growing controversy, are thought to play a role in synucleinopathies [15,16,17].

In addition to toxin-based and genetic models of PD, models based on injections of aggregated forms of aSyn have been developed. For this, monomeric recombinant aSyn is incubated in a well-defined manner to generate aggregated species consisting primarily of aSyn amyloid fibrils. These pre-formed fibrils (PFFs) are sonicated to produce shorter fibrils, enabling penetration of cell membranes and access to the cytosol. PFFs seed the aggregation of endogenous aSyn and trigger PD-like pathology in cell models, including phosphorylation at serine 129, and cell death [18,19,20]. Injection of aSyn PFFs into the dorsal striatum affects dopamine release and causes neurodegeneration and behavioral deficits that mimic alterations found in PD [21,22]. PFF-based models have been implemented in several groups although it has become apparent that the consistent production of aSyn PFFs is a major challenge and varies across laboratories [23]. Acknowledging this, the Michael J Fox Foundation established a set of guidelines that should be taken into consideration when using the PFFs model [23]. aSyn oligomerization and aggregation under different conditions have been widely studied in vitro. However, despite tremendous efforts, unexplained variations between experiments, batches, and laboratories, with regards to aggregation propensity of aSyn, remain a problem in the field.

## 2. Putative Diagnostic Assays Based on aSyn Seeding and Aggregation

aSyn seeding activity is being studied as a putative biomarker with diagnostic value since it differs markedly between patients and control brain or CSF. Seeding activity is usually detected using either a real-time quaking-induced conversion (RT-QuIC) or protein misfolding cyclic amplification (PMCA). Both are seed amplification assays (SAAs), based on the observation that misfolded aSyn in patient biospecimens can induce conformational change and aggregation of recombinant aSyn in vitro, in analogy to what was previously observed with the prion protein in prion diseases [24,25]. This seeded reaction amplifies pathological aSyn in biospecimens, allowing for detection of minute amounts of seed. In research settings, both methods show a high predictive power for PD diagnosis, thus, offering exciting possibilities for molecular diagnosis using accessible peripheral tissue. CSF analysis of PD patients results in sensitivity of 85–100% and 82–100% specificity [24,26,27,28].

Nonetheless, a number of problems need to be overcome for in vitro seeding assays to become routine screening techniques. The non-linear nature of the seed amplification is key for its high sensitivity, enabling the identification of minute amounts of aSyn seeds. On the other hand, slight variations in the assay conditions, including the substrate, can have a large impact on the assay kinetics. Besides the need to standardize protocols and sampling sites, production and purification of the recombinant aSyn as a substrate need to be standardized. Some studies use home-made aSyn, others use protein from commercial vendors. While batch-to-batch variations in aSyn are recognized as a possible source of variation in the field [28,29], only a subset of studies reports systematic comparison of inter-batch variability [28,30,31,32]. In a recent study, SAAs were performed on control- and PD CSF by three independent groups in parallel according to their respective protocols [33]. The study confirmed the high diagnostic value of SAAs: PD was diagnosed with a sensitivity of 86–96% and specificity of 93–100% but reported high variability with regards to assay parameters (AUC maximum fluorescence, time to threshold) measured. A more robust measurement of these parameters could potentially enable correlation with clinical features, further increasing the diagnostic power of SAAs.

## 3. aSyn Is Intrinsically Disordered

Conformational changes and aggregation of aSyn are thought to be associated with PD and related synucleinopathies and have, therefore, been a subject of intense research [34]. As stated above, monomeric aSyn is viewed as an intrinsically disordered protein (IDP). Under physiological conditions, a number of studies showed that aSyn, in its free state, is present primarily as natively unfolded protein in different cells [10,35,36]. In addition, aSyn produced in *E. coli* and purified under denaturing or nondenaturing conditions is an unstructured monomer [4]. During the aggregation process, aSyn molecules cluster together into oligomeric species that can differ in size and shape. At least some of these oligomeric species (on pathway) can then rapidly assemble into larger beta-sheet rich fibrillar structures with amyloid properties. Interestingly, recent data suggest that these fibrillar amyloid structures appear to have different arrangements in different synucleinopathies, in different disease subtypes, or even between patients. Detailed studies revealed that aSyn preparations with distinct morphologies lead to different pathologies and the individual morphological characteristics of aSyn strains can be amplified and propagated in vivo and in vitro [13,15,16,17,37]. The fibril morphology of aSyn strains depends on a variety of parameters, including pH and salt content in the buffer used [14]. In addition, it was recently shown that even if aSyn of the same batch is set to aggregate under identical conditions, ThT negative fibrils may emerge along with ThT positive fibrils, in a random equilibrium [38].

## 4. Varying Protocols Are in Use to Produce Recombinant aSyn

aSyn used for in vitro studies has typically been produced in E. coli and purified using different protocols (Table 1). aSyn isolation protocols differ mainly in terms of protein expression, extraction, purification, and storage. Recombinant aSyn expression in E. coli is the first step in protein preparation. Before inducing protein expression, a starter culture is used to inoculate larger cultures volumes. Expression of aSyn from inducible promotors is initiated using IPTG. Although several labs induce protein expression for four hours, others induce for shorter periods or up to 16 h. In addition, the IPTG concentration used varies and, in some protocols, it is not used at all, or not clearly stated. However, the induction time and/or IPTG concentration used likely affect the protein yield. Importantly, how these differences affect the dynamic ensembles of aSyn conformations and equilibrium has not been addressed.

Most differences among production protocols consider the second stage of protein preparation, i.e., aSyn extraction, which basically aims to separate the protein of interest from other unwanted bacterial proteins. Four extraction methods are commonly used for aSyn isolation. (1) One of the most commonly used methods is boiling (95–100 °C) the bacterial pellet suspension. Given that aSyn is a natively unfolded and relatively thermostable protein, boiling will lead to denaturation and precipitation of most heat sensitive unwanted bacterial debris. Centrifugation removes most of these proteins in the pellet and keeps aSyn in the supernatant for the next purification steps. Boiling time and extraction buffers vary among several labs. Most laboratories use heat treatments for bacterial cell lysates for 5–15 min, others for 20 or even 30 min. Although aSyn is considered relatively thermostable, heat treatment could result in partial protein degradation with a longer boiling time [39]. The composition of the extraction buffer is considered important and can affect interactions of the protein of interest with other cells structures such as bacterial nucleic acids. Some groups use Tris buffer as the main extraction buffer, while many other groups include also salt (sometimes high salt) in the extraction buffer.

aSyn can also be isolated through (2) ammonium sulfate precipitation, (3) acid precipitation, or (4) periplasmic lysis. Furthermore, some purification protocols include a combination or more than one method as summarized in Table 1.

**Table 1 biomolecules-12-00324-t001:** Overview over different protocols currently used for preparation and purification of recombinant aSyn protein.

aSyn Expression	aSyn Extraction	Purification	Storage	Refs.
LB medium, 1 mM IPTG, 4 h induction	40 mM Tris acetate buffer, sonication, and 10 min boiling	IEX, SEC	Lyophilized	[4]
TB medium, no IPTG, ON induction	High salt buffer, sonication, and 15 min boiling	SEC, IEX	Frozen	[19]
LB medium, 1 mM IPTG, 5 h induction	Periplasmic lysis by osmotic shock buffer	IEX	Lyophilized	[40]
LB medium, 0.5 mM IPTG, ON induction at 25 °C	10 mM Tris lysis buffer, 30 min boiling, and ammonium sulfate precipitation	IEX, SEC	Frozen	[37]
IPTG, other details unstated	Cell lysate precipitated in ammonium sulfate	IEX, SEC	Lyophilized	[41]
Details not stated	20 mM Tris and 100 mm NaCl, acid precipitation	IEX	Not stated	[42]
LB medium, 1 mM IPTG, 4 h induction	Periplasmic lysis by osmotic shock buffer	IEX, HIC	Lyophilized	[43]
LB medium, 1 mM IPTG, 4 h induction	20 mM Tris, sonication and 15 min boiling	IEX	Lyophilized	[44]
LB medium, IPTG concentration not stated, 4 h induction	10 mM Tris, freeze–thaw, sonication, 20 min boiling, and ammonium sulfate precipitation	IEX	Not stated	[14]
LB medium, 0.5 mM IPTG, ON induction at 25 °C	500 mM NaCl and 100 mm Tris, 10 min boiling, and acid precipitation	IEX	Not stated	[45]
LB medium, 0.44 mM IPTG, 2 h induction at 37 °C	10 mM Tris, freeze–thaw, sonication, and ammonium sulfate precipitation	IEX, SEC	Not stated	[46]
TB medium, 0.5 mM IPTG induction for 16 h at 37	100 Mm tris and 500 mm NaCl, freeze–thaw, 10 min boiling, and acid precipitation	RP-HPLC	Lyophilized	[47]
LB medium at 37 °C, 4 h IPTG induction	Sonication, 5 min boiling	IEX, SEC, RP-HPLC	Lyophilized	[48]
M9 medium at 37 °C, 1 mM IPTG induction for 4–5 h	10 mM tris, sonication, 20 min boiling, and ammonium sulfate precipitation	IEX, SEC	Frozen	[38]

During purification, the 3rd phase of aSyn production, different chromatographic methods are employed to ensure a protein of high purity. Although aSyn is frequently purified in two successive chromatographic steps of ion-exchange chromatography (IEX) and size exclusion chromatography (SEC), it is purified many times by one chromatographic step of IEX. Furthermore, some research groups use SEC as the first purification step followed by IEX. Moreover, a third purification step of Reversed-Phase High-Pressure Liquid Chromatography (RP-HPLC) is used in some studies. Usually, two chromatographic steps are used to make sure aSyn prepared is monomeric and does not contain oligomeric or high ordered species. This variation in the number and/or order of purification steps will most likely affect the percentage of purity of the protein.

The way purified aSyn is handled and stored is of very high importance as it can affect the protein stability in terms of degradation or aggregation. After finishing all the isolation and purification steps, aSyn is either stored frozen or lyophilized. Lyophilization allows for longer storage but was shown to greatly affect the monomeric structure and variability in ThT-based aggregation assays [49].

In summary, aSyn purification methods in use vary in all stages of aSyn purification. aSyn purified with these methods is frequently used to generate PFFs or oligomeric species. Although aSyn is defined as a natively unfolded protein, it is of high importance to keep in mind that it exists as a dynamic ensemble of conformations. Some of these conformations are more aggregation prone than others. These conformations are affected by the surrounding environment which can affect the equilibrium among this dynamic ensemble of conformations [50]. It is still largely unknown how the different purification methods, and how variations within a certain method can affect the final equilibrium of the different conformations. We think that variations, even minor changes, can affect the protein surrounding environment and subsequently the distribution of the different conformations of monomeric protein leading to different aggregation propensities. A recently published systematic comparison of aSyn isolated via boiling, acid precipitation, ammonium sulfate precipitation, or periplasmic lysis revealed that the aggregation propensity is affected by the isolation method [51]. In this comparative study, the authors observed that aSyn extracted by periplasmic lysis and ammonium sulfate precipitation has a higher aggregation propensity, compared to aSyn extracted by boiling, or acid precipitation. However, so far, no study has systematically compared the impact of shorter (5–15 min) heat-treatments that are used in most protocols [4,19,44,47,51,52]. In addition, different buffers are used to lyse the bacterial cells [4,17,19,37,44,47,51], and their effects are largely unknown.

## 5. Variations in Purification Protocols Can Change the Outcome of Downstream Applications

We provide here a practical example of how the lysate extraction buffer composition can affect the aggregation propensity of aSyn. While some protocols use a lysis buffer lacking sodium chloride (NaCl) salt, others include up to 750 mM NaCl. We lysed bacterial cells either in high salt buffer or no salt extraction buffer and compared the aggregation kinetics of aSyn after anion exchange chromatography (AEX) based on Thioflavin T (ThT) fluorescence sensitivity in 96-well plates. Importantly, before running the aggregation assays, we tested the purity of aSyn. Using sodium dodecyl sulphate–polyacrylamide gel electrophoresis (SDS-PAGE) and Coomassie staining, we detected only one band migrating above 15 kDa, corresponding to the monomeric state of the protein (Figure 1a). Before lyophilizing the protein purified by AEX, we analyzed an aliquot of each condition by size exclusion chromatography and detected mainly one peak in the gel filtration chromatogram, suggesting aSyn was essentially monomeric (Figure 1b). Shoulders just eluting before the monomeric fraction usually indicate the presence of multimers, here around 12 mL of elution volume. We also detected a small peak at 19 mL of elution volume in the “No salt” extracted aSyn. Although no protein was detected in this peak by SDS-PAGE (data not shown here), most likely due to very low abundance, we cannot rule out that a degradation product of aSyn is present, and that this may affect the aggregation. This could mean that salt might play a role in stabilizing aSyn during its extraction and purification. Detailed mass spectrometry analyses of the second peak might help further clarify this. Furthermore, it is important to keep in mind that, although ThT has been widely used to study protein aggregation and amyloid formation, recent studies showed that ThT reactivity is not necessarily a synonym of fibrilization or aggregation, as some types of fibrils are ThT-negative [38].

## 6. Collaborative Efforts Will Help with Comparing and Standardizing aSyn Production

Given that recombinantly produced aSyn is widely used for modeling PD-related pathology in cell- and animal models, and for in vitro studies of aSyn biology, a great deal of attention has been directed towards the preparation and characterization of aSyn PFFs species that are used for subsequent applications. The Michael J Fox Foundation has put forward suggestions for a standard protocol for production and thorough characterization of these species. Therefore, since (i) the morphology of aSyn aggregates is sensitive to environmental parameters including pH, biomolecules, and salt, and (ii) distinct strain characteristics seem to have the ability of being propagated, monomeric aSyn must be produced with great care and under standardized conditions in order to ensure aSyn quality and purity, and to reduce inter-laboratory variability that complicates the interpretation of findings (Figure 2). Therefore, we recommend true collaborative efforts between different laboratories in order to (i) systematically compare the impact of key steps in the production and purification protocols used and (ii) to develop a more general protocol with minimal variations in order to improve aSyn research, enabling the community to investigate both the physiology and pathophysiology of aSyn.

## 7. Materials and Methods

### 7.1. Expression and Purification of aSyn

Plasmid pET21-aSyn, containing human aSyn cDNA was transformed into competent *E. coli* BL21-DE3 (Sigma) on ampicillin agar plates, and one colony was used to create the bacterial cultures. Cultures of *E. coli* in 2x LB medium containing ampicillin (200 μg/mL) were grown at 37°C with shaking. Expression was induced for 2 h at OD_600_ 0.5–0.6 with 1 mM of isopropylβ-thiogalactopyranoside (IPTG). Bacteria were pelleted by centrifugation at 6600g for 15 min and lysed on ice in either “high salt buffer” (10 mM Tris pH7.6, 750 mM NaCl, 1 mM EDTA with protease inhibitor (cOmplete, Roche)) or “No salt buffer” (10 mM Tris pH7.6, 750 mM NaCl, 1 mM EDTA with protease inhibitor). Lysates were sonicated on ice for a total of 5 min (30 s on, 30 s off pulses, 60% power), and boiled for 15 min at 95 °C. Boiled protein samples were then centrifuged, and the supernatant containing aSyn was collected. After dialysis into 10 mM TRIS pH 7.6, 1 mM EDTA, 50 mM NaCl, supernatants were purified with anion exchange chromatography (HiTrap Q HP, GE Healthcare) using a mobile phase of 25 mM Tris pH 7.6 and a linear gradient of 9 column volumes of elution buffer to 1M NaCl on an Äkta Pure 25M (Cytiva). “No salt” extracted protein sample was loaded directly to the anion exchange column without dialysis. Fractions containing pure aSyn were identified on a Coomassie-stained SDS-PAGE, pooled, and extensively dialyzed into water. Following lyophilization (Zirbus), protein was stored at −20 until further use. aSyn concentration was estimated by measuring protein absorbance at 280 nm using the molar extinction coefficient 5960 M^–1^ cm^–1^. Protein purity was checked by running 5 ug of each sample on 15% SDS-PAGE gel and staining the gel with Coomassie dye. Before lyophilization, 500 µL of each condition were loaded to superdex 200 column (Cytiva) to further check the protein purity.

### 7.2. Thioflavin-T-Based Aggregation Assays

aSyn was dissolved in 50 mM HEPES (pH 7.4), 100 mM NaCl, filtered through 0.2 um Spin Filters (Corning Incorporated), and centrifuged at 20,000× *g* for 20 min at 4 °C to remove any possible insoluble aggregates. Then, 10 mM ThT (Sigma-Aldrich, St. Louis, MO, USA) was added to 100 μL of aSyn (1 mg/mL). Samples were loaded onto black 96-well plates (Corning Incorporated, Seattle, WA, USA)) and covered with sealing tape. Plates were incubated at 42 °C with one minute orbital shaking at 432 rpm, and a 3.6 min waiting period for 985 cycles. Fluorescence was measured every cycle using a Tecan plate reader (Infinite M200 fluorescence plate reader, TECAN). Excitation was set at 440 nm with 20 flashes, and the ThT fluorescence intensity was measured at 480 nm emission, a 100 gain value. Four wells were used per condition. Data were normalized to the sample with the maximum fluorescence intensity for each plate.

## Figures and Tables

**Figure 1 biomolecules-12-00324-f001:**
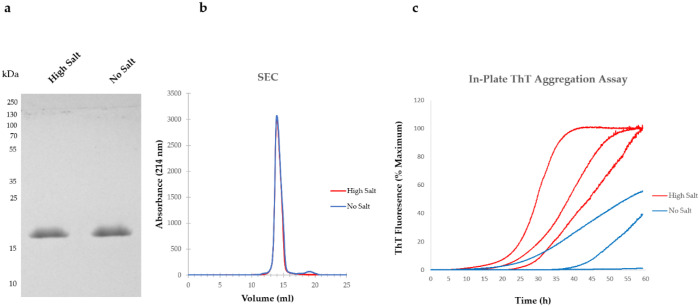
Effect of cell lysis buffer on the aSyn aggregation assays. (**a**) aSyn purity after AEX checked by SDS-PAGE and Coomassie staining. A total of 5 µg of AEX-purified aSyn from both extraction methods were loaded to the gel. (**b**) SEC chromatogram of aSyn after AEX. (**c**) Aggregation profiles of aSyn purified using differently extracted cell lysates monitored by changes in ThT fluorescence over time. *N* = 3 and each *N* represents an individually produced aSyn batch.

**Figure 2 biomolecules-12-00324-f002:**
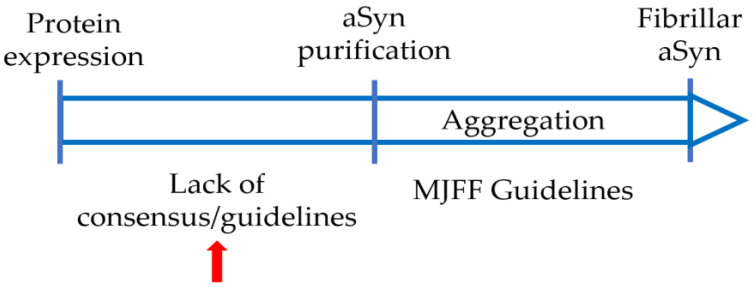
A schematic summary of the process of aSyn expression, purification, and aggregation. The red arrow points at the process of aSyn purification, where there is a lack of consensus, which is the focus of this study.

## Data Availability

Not applicable.

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
