# Peer review of "Production of Recombinant Alpha-Synuclein: Still No Standardized Protocol in Sight"

_biomolecules, 2022, doi:10.3390/biom12020324_

Round 1

Reviewer 1 Report

Al-Azzani et al. summarize and compare the variety of protocols for the production of recombinant alpha-Synuclein (aSyn) and their potential impact on the aggregation properties, which is a very essential and relevant issue.

Currently, the article focuses mainly on the importance of a standardized protocol for the production of pre-formed fibrils (PFFs) for seeding experiments in cell culture and animal models. However, another increasingly important application of recombinant aSyn for which reproducibility and comparability between batches and laboratories is highly relevant are in vitro seeding assays RT-QuIC or PMCA, respectively, which aim for reliable and precise intra vitam diagnoses of various neurodegenerative diseases, including synucleinopathies. Therefore, I wonder if the authors could point out and discuss this topic in addition.

Review articles should ideally cite and reference the original literature to support certain conclusions, statements etc. Already the first citations include two recent review articles, which cite again other articles. This needs to be thoroughly revised throughout the entire manuscript.

Except for the subheading „1. Introduction“, there is no obvious structuring. I assume, that the other subheadings have just been lost.

Minor issues:

  1. 2, line 72: … unwanted bacterial à proteins. …
  2. 3, line 4: “does not” instead of “doesn’t
  3. 3, line 108: 2x “conformation” instead of “confirmation
  4. 4, line 123: missing space: 750 mM
  5. 4, line 124/125: brackets must be removed

Reviewer 2 Report

The authors highlight issues in an important area in PD research, how methods to isolate recombinant aSyn can affect aggregation rate, fibril formation and potentially relevance to disease. Here they highlight that the buffer that aSyn is isolated in, high salt (750 mM NaCl) and no salt, could affect the aggregation rate in subsequent downstream assays.

Major comments

  • Although the authors have shown that the aggregation rate is different with salt and no salt in the isolation method, they haven’t characterised the starting material properly to show where this variability originates. The gel in Figure 1a I assume is a denaturing gel, but there are no methods to check this, may not show presence of dimers or oligomers in the samples. Usually, SEC is performed to ensure isolation of monomeric protein, yet it doesn’t seem clear if this was done. Before making claims that the isolation buffer impacts the aggregation rate the starting aSyn material needs to be characterised better to determine whether there are differences in proportions of monomer/dimer/oligomer or at least determine differences in the aSyn states between salt and no salt isolations. At a minimum I would expect at least one experiment to show this, e.g. SEC, native gel, DLS, native MS. I don’t think the paper is useful or can make claims on aggregation rates without an experiment to show the state of the starting aSyn material.

  • The whole paper is based on the importance of method choice to isolate aSyn, yet there are no methods in the draft I have received.

What AEX column was used, was figure 1a) a denaturing gel, what concentration of aSyn and ThT was used in the aggregation assay, what buffer was used in the aggregation assay, how many wells per aggregation assay were used? Etc.

Minor comments

More detail could be added in the introduction as currently these sentences are a bit ambiguous,  lines 29, ‘in other tissues’ line 32, ‘different conformations’, line 35-36 ‘highly heterogeneous’

Page 2 line 50, what is meant by ‘aSyn transformations’?

Page 2 line 72, missing a word ‘heat sensitive bacterial…’

Page 3 line 102, incorrectly referenced. Lyophilisation is known to induce oligomerisation, this paper shows that even after SEC of lyophilized samples there are still differences in aSyn aggregation rate and variability and oligomer shapes.

Page 3 line 117-118, the authors mention impact of short heat treatments, but do not investigate this in the paper, is this important?

Page 4 line 123, ‘lysed’ - lysed the cells how, heat?

Figure 1.  the legend could have more detail and the x axis on 1b has very strange numbering

Reviewer 3 Report

In this article, Al-Azzani and colleagues pinpoint the need of a standardized protocol for the production, extraction and purification of recombinant alpha-synuclein in modelling synucleinopathies using synthetic amyloids. Authors make a very short review of the state-of-the-art of procedures used in this aim, and a simple experiment for illustrating their message.

While I find the subject of a tremendous importance in the field, I raised several points that need to be addressed, developed, or implemented. In summary, I have no problem with a very short communication about this caveat, but it then needs to be flawless.

Major points :

1 - Regarding their experiment for showing the effect of cell lysis buffer :

The authors write an article about the importance of standardizing aSyn production and fibrillization procedures but the information about this experiment is extremely sparse:

What concentration ? fibrillization method ? Starting batch was purely monomeric ? What fibrillization buffer ? etc …

This latter point (fibrillization buffer) is of extreme importance : if they used no-salt buffer for fibrillization : they simply made Ribbons that are ThT low (Bousset et al 2015) ; if they used high salt everywhere, then it is another very interesting story…).

Also, authors should not talk about “fibrillization curves”, as they only measured ThT fluorescence : it is well known now that some ThT negative fibrils might be produced in these conditions. Talking about ThT negative amyloids, authors did not discuss anywhere the article by De Giorgi et al, Science Advances, 2020, while the latter even describes the spontaneous generation of different polymorphs (ThT+ and -) from a single production batch, worth including their procedures in Table 1.

2- The authors discuss only the importance of standardization of production procedures for PFF formation, while another topic is based even more on aSyn monomeric features : PMCA and RT-QuiC used as diagnostic tools ! These protocols can make the results differ even from one lab or one batch to another ! Authors should thoroughly include this in the discussion.

3- There is only an Introduction part, authors should include subheadings for the other parts.

Minor points :

l.16 : Abstract : typo “synucleiOnopathies”

l.72 : “bacterial” lacks “material” or else

l.97 : should be “The way purified aSyn”

l.97 - l.100 : the two senteces say exactly the opposite meaning “is of very high importance” and “will not have significant effects”… Also, freezing aSyn was shown to have an impact on fibillization.

l.139 : typo “amyoid”

Reviewer 4 Report

The paper presented by Al-Azzani  et al. is focused on the production and the purification of recombinant aSyn which is a crucial point concerning the entire community interested in synucleinopthies.  It provides an overview of the techniques used in various laboratories to produce and purify a-syn which is the prime material for all aggregation experiments. The authors point out the question of reproducibility and reliability among batches and laboratories in all the studies based on experimental synucleinopathy models.

Indeed the use of recombinant  aSyn is central for most of the works in the domain: it is used for producing PFFs which are the only well-known inducer of synuclein aggregation in cells and animals; but also in biophysical studies to understand the molecular dynamics of the aggregation process in vitro, i.e.  monitoring amyloid formation with fluorescent probes; and but not least, in the amplification of a-syn extracted from tissues or fluids from patients (PMCA, RTquic) to develop diagnostic tools. 

The article is structured in two parts: the first one is a reviewing work describing the different steps involved in a-syn production and purification. The second is an experimental part providing an example of the impact of a specific step of a-syn production (lysis extraction buffer) on the ability of a-syn to self-aggregate and/or to bind THT. In this respect, it should be reminded that a damped or even null ThT signal is not synonym with an absence of aggregation and can hide the presence of fibrils.

General comment: the molecular rationale on the possible impact of the production & purification procedures on the outcome of a-syn aggregation experiments is only explained at the end of the paper. This prevents the reader from being really interested in the first part. Indeed, as very little work has been published to experimentally document whether the procedures of production of recombinant a-syn could induce a variability in its aggregation propensity, the authors only compile a list of different existing protocols, mentioning in a declarative manner that they could be a source of variability. However, to deliver a more impactful message, I think that the article would benefit from being rewritten with the aim of stating and explaining the molecular basis of the problem beforehand. It would be appropriate to introduce the concept of IDP, and the theoretical possibility that the environment favors some specific conformations of monomers over others and could influence the aggregation kinetics or force the acquisition of a specific amyloid conformation. These points (kinetics of aggregation and amyloid polymorphisms) are currently considered to be markers of, or to play a causal role in synucleinopthies. In particular it would be interesting to give more details on the results of the few comparative works published in the litterature and not just to cite them.

 Some more detailed points:

 1) expression in E. Coli: some labs use tagged forms of a-syn. The presence of the tag could influence the conformation. In the context, this is an important point to be considered.

 A-syn can aggregate spontaneously. Is there evidence of intra bacterial aggregation linked to its high concentration as for other recombinant proteins ?

 2) lysis & extraction: this part should be better detailed. It seems to me that some protocols are indeed simple lysis protocols and the purification is done by the SEC/ IEX stage. Other procedures allow to obtain a first purification of a-syn already before SEC (heat, precipitation). The principles of these protocols should be better explained in order to be able to assess their impact on a-syn.

Part 2: Authors provides an example of the variability induced by the extraction buffer on aggregation kinetics. Considering the context and the message of this work which is is to invite colleagues  to standardize protocols, it is rather incongrous that there is not a rigorous description of the Material & method relative to the single experiment shown in the paper.

 Final comment:

 The authors’ intention is to highlight the need for standardizing procedures of a-syn production & purification. I think this is indeed particularly important in order to be able to yield inter-lab consistency in the field and especially to develop diagnostic tools based on rtquic for example. On the other hand before standardization, I think it would be suitable also to encourage to a comparative work on the impact of specific key steps  and to obtain an experimental demonstration that the conformations of monomers favored by the environment could have consequences on the conformation of the resulting amyloid or on the aggregation dynamics. This could not only improve experimental procedures but also increase the knowledge of a syn biology.

The point raised by the article is new and fundamental for the improvement of studies in the field. For this reason, the message should be rewritten in a solid and impactful way to encourage researchers to work in a controlled manner and to understand the molecular basis of variability.

Ligne 107-108 confirmation instead of conformation

Round 2

Reviewer 1 Report

The authors have sufficiently addressed all my issues.

Author Response

We want to thank the reviewer for careful revision. We are pleased the reviewer  was satisfied with our revised version

Reviewer 2 Report

I'm going to nitpick because I think the topic the authors are discussing is really important for the field and therefore the results need to be described and presented in as much detail/clarity as possible. 

They have added a chromatograph of the protein elution from the SEC column, but have neglected to comment on the second peak at 19 mL in the 'no salt' condition. This could be an aSyn degradation product, or other contaminant, which subsequently affects the aggregation kinetics seen in the ThT-based assays. Could the authors rule out that this is a protein contaminant that can't be detected on SDS-PAGE due to abundance by using either mass spec or analysis of the peak using different wavelengths?

Additionally, there is also a slight shoulder in the 'no salt' condition at 12 mL just before the peak of the monomer. This shoulder is usually due the presence of multimers, which again is not mentioned or investigated further. 

It feels like the authors have interpreted the chromatograph to fit their narrative, rather than trying to investigate the origin of the difference in aSyn aggregation rate in the ThT-based assay. The results of the SEC data suggest that salt is needed to stabilise aSyn during lysis and purification, or at the very least could be interpreted as to provide a more monomeric, less contaminated product?

Furthermore, in the conclusion it mention that the MJFF have given guidelines for producing aSyn, how does their data fit with these guidelines? Does MJFF suggest the use of salt?

Are the ThT-based assays reflective of the amount of aSyn aggregated? The authors could run remaining monomer experiments e.g. SDS-PAGE or SEC to determine if the ThT fluorescence is reflective of the aggregation state of aSyn. 

The authors have now added their methods, but I am unclear why the ThT-based assay was carried out at 42C, this seems high. 
